# Predictive Clinical and Biological Criteria for Gene Panel Positivity in Suspected Inherited Autoinflammatory Diseases: Insights from a Case–Control Study

**DOI:** 10.3390/genes14101939

**Published:** 2023-10-14

**Authors:** Lionel Heiser, Martin Broly, Cécile Rittore, Isabelle Touitou, Sophie Georgin-Lavialle, Guilaine Boursier

**Affiliations:** 1Laboratoire de Génétique des Maladies Rares et Autoinflammatoires, Service de Génétique Moléculaire et Cytogénomique, National Reference Center for Autoinflammatory Diseases and AA Amyloidosis, Centre Hospitalier Universitaire Montpellier, Université de Montpellier, 34295 Montpellier, France; lionel.heiser@etu.umontpellier.fr (L.H.); martin.broly@chu-montpellier.fr (M.B.); c-rittore@chu-montpellier.fr (C.R.); isabelle.touitou@inserm.fr (I.T.); 2Stem Cells, Cellular Plasticity, Regenerative Medicine and Immunotherapies, INSERM, 34295 Montpellier, France; 3Tenon Hospital, Centre de Référence des Maladies Auto-Inflammatoires et des Amyloses Inflammatoire (CEREMAIA), Internal Medicine Department, Sorbonne University, AP-HP, 4 rue de la Chine, 75020 Paris, France; sophie.georgin-lavialle@aphp.fr

**Keywords:** hereditary recurrent fevers, high-throughput sequencing, inflammation, predictive value of tests, retrospective studies

## Abstract

In order to assess the clinical and biological criteria that predict gene panel positivity in patients with a suspected inherited genetic autoinflammatory disease, we conducted a case–control study. These new selection criteria could replace the national multidisciplinary staff approval before performing genetic testing that has been required since 2019. The study involved 119 positive gene panels matched by panel sizes to 119 randomly selected negative gene panels. The patients were referred to our laboratory for genetic testing between June 2012, and March 2023. The clinical and biological criteria were extracted from a prospectively filled database. We focused our evaluation on accuracy and the positive predictive value. Neonatal symptom onset and deafness had the highest accuracies among all criteria associated with the positivity panel, with 92.9% (88.6; 96.0) and 92.6% (88.5; 95.6), respectively. However, it is important to note that the associated Positive Predictive Values (PPVs) cannot exceed 50%. Despite finding a statistical association between clinical and biological criteria and panel positivity, the predictive values of these criteria were not sufficient to recommend Next-Generation Sequencing (NGS) gene panel testing without the national multidisciplinary staff evaluation.

## 1. Introduction

Autoinflammatory diseases are a subset of immune-related disorders specifically arising from innate immune system hyperactivation. The most typical symptoms associated with these diseases are recurrent fevers, arthralgia, skin rash, or biological inflammation marker concentration elevation, such as C-Reactive Protein (CRP) or Serum Amyloid A protein (SAA). Knowing these diseases are influenced by genetic and environmental factors, their pathophysiology is complex [1]. One of the first monogenic autoinflammatory disease to be studied was Familial Mediterranean Fever (FMF, OMIM #249100) for which Mendelian inheritability was described for the first time in the Armenian population in 1989 [2]. The molecular basis of this disorder was described in 1997 [3,4] as a mainly autosomal recessive condition.

Since 1997, sequencing technique advances associated with more and more efficient technology allow faster, cheaper, and more precise diagnoses [5]. Following this trend, Next-Generation Sequencing (NGS) gene panels have increased in size to consider new gene discoveries. In our laboratory, we developed a unique NGS panel, which investigates inherited autoinflammatory diseases. The panel has undergone expansion in terms of the number of genes over time (32 to 302 genes). In the meantime, requests for NGS testing of patients have increased by orders of magnitude as well as the laboratory workload making it is almost unmanageable to return results in a reasonable timeframe, not to mention the low diagnostic rate [6]. In order to better select patients for whom the expected benefit of genetic testing would be the greatest, a national multidisciplinary staff approval before performing genetic testing has been required since 2019 [6]. This process requires important human resources, which is why we want to move on to implementing new selection criteria. Bypassing this national staff would result in several logical consequences, including fewer requests, more time for difficult cases, and possible other resource savings. Therefore, the objective of this study is to evaluate the performance of the use of clinical and biological criteria to predict gene panel testing positivity.

## 2. Methods

### 2.1. Selection of Cases and Controls

We used positive gene panels as cases. A positive panel is defined as a panel for which at least one pathogenic (P), likely pathogenic (LP), or Variant of Unknown Significance with hypothetical effect (VUS+) was found according to the ACMG variant classification guidelines [7]. The differentiation between Variant of Unknown Significance (VUS) and VUS + was at the geneticist’s discretion.

We considered a gene panel to be positive when biallelic variants were found for autosomal recessive diseases or X-linked recessive diseases in females and when at least one heterozygous variant or one hemizygous variant was found for autosomal or X-linked dominant diseases or X-linked recessive disorders in males, respectively. Negative panels were defined by all other situations consisting of the discovery of predisposition factors that were unable to explain the patient’s complete symptomatology.

### 2.2. Clinical and Biological Characteristics

Clinical and biological characteristics were collected by the prescriber as a questionnaire exploring, among other items, duration of the symptoms, age of onset, crises characteristics, CRP values during crises, clinical symptoms, and other relevant medical history, biological test results (mevalonic aciduria levels, autoantibodies, blood cell counts, etc.), prior genetic test results, and treatment information.

These data were prospectively added into our laboratory information management system database. Clinical and biological information was manually curated from archived paper files if absent in our database.

Missing clinical or biological qualitative characteristic information in the questionnaire was considered as negative except for fever and CRP elevation, which were considered negative in cases and positive in controls even in the absence of information from the questionnaire. Missing age-of-onset data were excluded.

### 2.3. Study Design

All definitive panel results for samples received between September 2014 and June 2023 were extracted from our in-house database. All positive panels constituted the case group (*n* = 119). We randomly selected the negative control panels in a 1:1 ratio using a random drawing process. Because our gene panel composition evolved over time to reflect new gene discoveries, matching was performed taking into account specific panel versions. Therefore, cases and controls were matched by panel size (32, 55, 62, 114, and 302 genes).

### 2.4. Statistical Analysis

Statistical analysis was performed using a Student’s *t*-test for continuous variables (age of onset) and χ^2^ test or Fisher’s exact test for qualitative or dichotomous variables. Predictive performances (sensitivity, specificity, positive and negative predictive values, and accuracy) and their respective confidence intervals at 95% were only estimated for significant associations except for age of onset, fever, and CRP elevation because these parameters are most of the time the reason for genetic testing.

Subgroup analysis was performed to explore the impact of time (associated with the evolution of panel size) on these associations and trends. Exploratory analysis was designed by excluding all panels with missing data about fever and/or CRP elevation.

Significance level for all analyses is given by a *p*-value < 0.05. A trend is defined as an association with a *p*-value ≥ 0.05 and <0.10.

### 2.5. Patients

Written informed consent was obtained from the patients or their parents in the case of minor patients before ordering genetic testing.

## 3. Results

### 3.1. Positive Panel Results

The case group comprised 119 positive samples for which prescriptions were issued between 27 June 2012 and 23 March 2023. During this timeframe, around 1700 samples were returned, resulting in an estimated positivity rate of 7%. There was significant variability in the frequency of the observed gene variants (see Figure 1). For instance, we detected 26 variants in *MEFV* (constituting 17% of all identified variants), 22 in *MVK* (14%), and 20 in *NLRP3* (13%). These variants ranged from rare VUS, which might contribute to the phenotype, to pathogenic variants fully explaining the symptoms. In total, there were 154 variants across 119 patients.

### 3.2. Clinical and Biological Characteristics

Clinical and biological characteristics (Table 1) were available for all patients except one. Significant differences between controls and cases were age of onset, especially neonatal onset (12 vs. 1 in cases and controls, respectively. OR = 13.83 95% Confidence Interval (95% CI) (1.76; 108.38)), onset during early childhood (56 vs. 30 in cases and controls, respectively. OR = 2.99 95% CI (1.69; 5.30)), deafness (19 vs. 2 in cases and controls, respectively. OR = 11.12 95% CI (2.53; 48.89)), and failure to thrive (24 vs. 12 in cases and controls, respectively. OR = 2.25 95% CI (1.07; 4.75)). All of these associations were related to panel positivity. Conversely, pharyngitis was associated with panel negativity (14 vs. 26 in cases and controls, respectively. OR = 0.48 95% CI (0.24; 0.97)). Missing age-of-onset data concerned 15 cases and 12 controls.

Furthermore, we examined combinations of characteristics, particularly focusing on the number of systems involved (excluding the inflammation category), in order to delve into the multisystem nature of inflammatory diseases, as detailed in Table 2. Among these, only the presence of hepatosplenomegaly was linked to an increased risk in the case group (14 individuals, 11.8% in cases vs. 5 individuals, 4.2% in controls, with an odds ratio of 3.04 and a 95% CI (1.06; 8.73), *p*-value 0.03). There is no observed association related to the number of affected systems, and their distribution appears to be similar.

### 3.3. Diagnostic and Predictive Values

Table 3 presents the diagnostic and predictive values linked to characteristics associated with panel positivity, along with fever and elevated CRP during crises. Among these, two characteristics stand out with an estimated accuracy exceeding 90%: neonatal onset (92.9 95% CI (88.6; 96.0)) and deafness (92.6 95% CI (88.5; 95.6)). Furthermore, the sensitivity, specificity, and Positive and Negative Predictive Values (PPVs and PNVs) for neonatal onset are 11.5% (6.1; 19.3), 99.1% (94.9; 100), 48.2% (11.0; 87.5), and 93.7% (88.6; 96.0), respectively. For deafness, the corresponding values are 16.0% (9.9; 23.8), 98.3% (94.1; 99.8), 41.7% (14.6; 75.0), and 94.0% (93.8; 94.4). Despite a relative high sensitivity, the PPV and accuracy of fever and CRP elevation during crisis remained very low. Interestingly, the most sensitive indicators appear to have relatively low diagnostic accuracy, whereas the most specific characteristics (which are also significantly associated with panel positivity) demonstrate higher levels of accuracy.

### 3.4. Panel Subgroup Analyses

The subgroup analyses are detailed in Table 4. The distribution of panels within each case–control subgroup is as follows: 5 in the 32-gene subgroup, 17 in the 55-gene subgroup, 57 in the 62-gene subgroup, 9 in the 114-gene subgroup, and 31 in the 302-gene subgroup. Upon re-evaluation of the identified correlations and patterns within each matched group, these resurfaced predominantly in the largest group, which comprises 62 genes in the panel. Notably, correlations are observed for neonatal onset (16.3% in cases vs. 0% in controls, *p*-value 0.006), early childhood onset (61.2% in cases vs. 34.7% in controls, *p*-value 0.009), deafness (14% in cases vs. 1.8% in controls, *p*-value 0.03), and pharyngitis (12.3% in cases vs. 28.1% in controls, *p*-value 0.04). No association was found for failure to thrive or hepatosplenomegaly in the 62-gene panel subgroup or for any characteristic in any other subgroups. Additionally, within the 62-gene panel subgroup, bipolar aphthosis, which was a statistical trend in the principal analysis, seems to be linked to panel negativity in the subgroup analyses.

### 3.5. Exploratory Analyses

Exploratory analyses are presented in Table 5. After excluding missing data related to fever and elevated CRP during crises, no association was found for fever (OR 1.37 95% CI (0.74; 2.51), *p*-value 0.32) and CRP elevation during crisis became significantly associated with panel positivity (OR 7.76 95% CI (2.24; 26.88), *p*-value 0.0002).

## 4. Discussion

### 4.1. Clinical and Biological Criteria as Positive Panel Predictors?

The questionnaire given to the prescribing clinicians encompassed both clinical and biological criteria. Despite undergoing modifications over time, the core clinical indicators have remained consistent. To address this variation, we partially controlled by matching the panel sizes (32 genes, 55 genes, 62 genes, 114 genes, and 302 genes). This approach enabled us to not only match by time period but also account for differences in the questionnaires simultaneously. However, these clinical and biological criteria exhibited an uneven ability to distinguish a genetic pattern within a patient’s clinical presentation.

In fact, certain symptoms lack specificity for autoinflammatory diseases, such as pain-related symptoms (chest pain, headaches, arthralgia, abdominal pain, etc.) [8]. Conversely, some symptoms are overly specific and rare, making them inadequately represented in this study. Examples include cerebral calcifications [9] and myelodysplastic syndrome, which typically manifests in older patients but may appear in around 50% of patients with VEXAS syndrome, for whom a panel is not sequenced [10].

The absence of an association between fever and CRP elevation during crises with NGS panel positivity, along with the limited diagnostic and predictive values, might be attributed to the fact that these two indicators serve as warning signals for panel prescriptions in cases of unexplained situations. Consequently, they are highly prevalent in panel requests. Conversely, a patient, particularly an adult, without fever and/or CRP elevation would have a lower likelihood of undergoing genetic exploration. In contrast, the consideration of a genetic etiology becomes more apparent when dealing with a child and/or a severe clinical presentation, regardless of fever and/or CRP elevation.

It is important to interpret these findings with some caution, considering exploratory analyses that reveal a significant association between elevated CRP during crises and positive NGS panels when cases with missing CRP data are excluded. However, these alterations do not significantly modify the diagnostic values.

The statistically significant associations identified in our primary analyses are limited to a few clinical parameters. Among these, neonatal onset emerges as one of the most robust diagnostic and predictive values. For a patient exhibiting symptoms of neonatal onset suspected of genetic autoinflammatory diseases, there is a 50% chance of receiving a confirmed genetic diagnosis. Similarly, a patient with a positive panel has a 13 to 14 times higher likelihood of experiencing neonatal onset compared to an individual with a negative panel. Nonetheless, neonatal onset is not the predominant age of onset, as only a small number of patients in each group fall into this category. On the other hand, early childhood onset encompasses a greater number of patients displaying an association with positive panels. However, its diagnostic and predictive values are comparatively less accurate than neonatal onset, except for sensitivity, which is logically higher. Our findings align with the results of Similuk et al. [11], where youth was associated with positive molecular findings after exome sequencing of 1000 families, with a median age for individuals with positive molecular testing of 22.0 years, in contrast to 43.0 years for those with negative results.

Only neonatal onset and deafness came close to achieving a positive predictive value of 50%. Whether this suffices or not hinges on the risk threshold we are willing to accept to replacethe involvement of multidisciplinary teams. Unfortunately, the study did not identify any single characteristic or combination of characteristics robust enough to fulfill this role. We are investigating genetically and phenotypically distinct rare diseases. The low diagnostic yield (i.e., estimated at 7%) could contribute to the absence of obtaining clinical and biological criteria to predict gene panel testing positivity.

Regarding neonatal onset, its association with a more severe clinical presentation might warrant urgent genetic testing without the need for multidisciplinary approval. This dilemma prompted McCreary et al. to devise a workflow with a turnaround time within 48 h for such cases [12]. Concerning deafness, this symptom could potentially be part of a genetic syndrome, as seen in *NLRP3*-related syndromes [13], which happens to be one of the most frequently encountered genes in our case group.

Regarding the panel subgroup analyses, the predominant associations resurfaced exclusively within the 62-gene panel. This finding is likely attributed to the smaller sizes of the other subgroups. Additionally, it is noted that our panel subgroup analyses revealed a significant association between bipolar aphthosis and NGS panel negativity in the 62-gene subgroup. Pediatric Behçet’s disease is occasionally of monogenic etiology, such as when the *TNFAIP3* gene is mutated [14]. This gene was included in our panel since the 55-gene panel version. However, it is likely that other monogenic etiologies have not yet been identified.

### 4.2. Limitations

The study’s observational nature inherently exposes it to unaccounted confounding biases. However, we managed to mitigate some through matching based on panel size. It is possible that some genuinely positive panels, characterized by variants classified as VUS rather than VUS +, might have been overlooked. These panels, labeled as negative in our study, could potentially exhibit distinct clinical presentations. Nonetheless, we chose to prioritize a higher degree of certainty in directly linking variants to patient symptomatology by selecting only VUS+, Likely Pathogenic (LP), and Pathogenic (P) variants.

## 5. Conclusions and Perspectives

While our study reveals statistical associations between various clinical and biological criteria and panel positivity, the predictive values of these associations for patients suspected of genetic autoinflammatory diseases cannot readily be used for bypassing the multidisciplinary team’s decision-making processes due to insufficient diagnoses and predictive values. Further studies are necessary to identify the best possible criteria, especially specific symptoms and more complete biological characterization. However, these results highlight a challenge: we need to help prescribers to identify high-risk patients for genetic autoinflammatory diseases more easily and particularly to establish genetic diagnoses in adulthood, while it is known that diagnostic odyssey is prominent in autoinflammatory diseases [15].

A new approach could involve reducing gene panel size in the initial phase for situations where the symptomatology let us strongly suspect one or several genes, which would allow prescriptions without relying on multidisciplinary staff decisions. Thus, the time saved could be reallocated to more extensive investigations, such as larger panels, whole exome sequencing (WES), or whole genome sequencing (WGS). This strategy could be implemented in scenarios involving negative outcomes in the preliminary testing or under particular circumstances, like urgent diagnoses. This approach would be close to the guidelines of the International Society of Systemic Auto-Inflammatory Diseases (ISSAID) [16].

## Figures and Tables

**Figure 1 genes-14-01939-f001:**
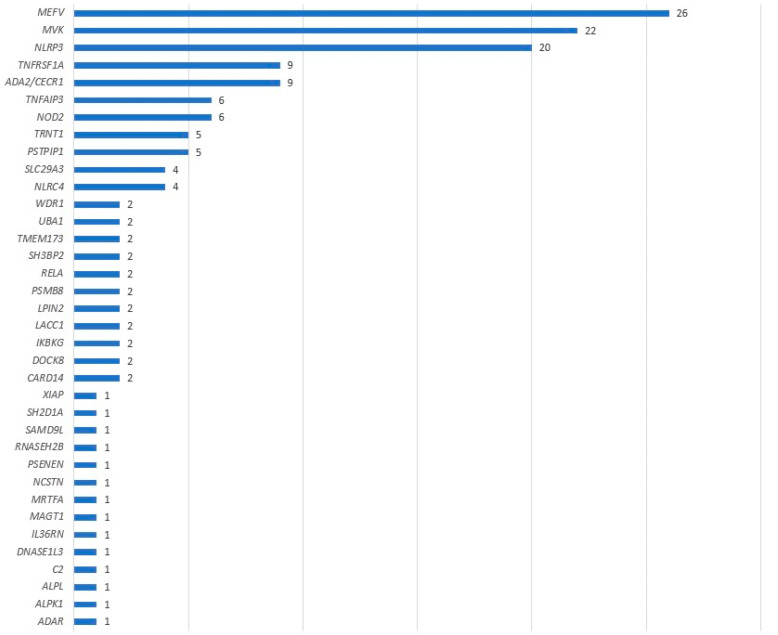
Number of causal variants identified by genes.

**Table 1 genes-14-01939-t001:** Patient characteristics at the time of prescription.

Characteristics	Cases *n* (%)	Controls *n* (%)	*p*	Characteristics	Cases *n* (%)	Controls *n* (%)	*p*
Age of onset				Digestive			
mean (months), 95% CI	96 (61; 131)	137 (106; 168)	0.09	abscess	2 (1.7)	4 (3.4)	0.68
**neonatal (<30 days old) ^§^**	**12/104 (11.5)**	**1/107 (0.9)**	**0.001**	hepatic cytolysis	11 (9.2)	14 (11.8)	0.53
**early childhood** **(≤3 years old) ^§^**	**56/104 (53.8)**	**30/107 (28.0)**	**0.0001**	IBD	2 (1.7)	6 (5.0)	0.28
childhood (<18 years old) ^§^	94/104 (90.4)	87/107 (81.3)	0.06	diarrhea/vomiting	34 (28.6)	34 (28.6)	1
adult (≥18 years old) ^§^	10/104 (9.6)	20/107 (18.7)	0.06	abdominal pain	47 (39.5)	52 (43.7)	0.51
Inflammation				hemorrhage	3 (2.5)	7 (5.9)	0.20
fever	86 (72.3)	87 (73.1)	0.88	Cutaneous			
crisis CRP elevation	102 (85.7)	98 (82.4)	0.48	aphthous ulceration *	42 (35.3)	42 (35.3)	1
Thoracic				erythema nodosum	4 (3.4)	7 (5.9)	0.34
chest pain	25 (21.0)	25 (21.0)	1	folliculitis, acne, HS	18 (15.1)	19 (16.0)	0.86
pleuro-pericarditis	9 (7.6)	9 (7.6)	1	lipodystrophia	4 (3.4)	4 (3.4)	1
pneumonia	8 (6.7)	8 (6.7)	1	livedo	6 (5.0)	6 (5.0)	1
Neurologic and sensorineural				maculopapular rash/urticaria	42 (35.3)	33 (27.7)	0.21
stroke	3 (2.5)	7 (5.9)	0.20	necrosis	4 (3.4)	6 (5.0)	0.52
cerebral calcifications	2 (1.7)	1 (0.8)	1	edema	8 (6.7)	9 (7.6)	0.80
headaches	26 (21.8)	36 (30.3)	0.14	pseudoerysipelas	9 (7.6)	7 (5.9)	0.60
conjunctivitis	21 (17.6)	13 (10.9)	0.14	psoriasis	6 (5.0)	4 (3.4)	0.52
encephalitis	1 (0.8)	2 (1.7)	1	pyoderma	0 (0.0)	3 (2.5)	0.25
epilepsy	1 (0.8)	3 (2.5)	0.62	tenosynovitis	6 (5.0)	6 (5.0)	1
meningitis	8 (6.7)	3 (2.5)	0.12	vasculitis/Raynaud	14 (11.8)	10 (8.4)	0.39
papillitis	5 (4.2)	0 (0.0)	0.06	Diverse			
uveitis	11 (9.2)	9 (7.6)	0.64	**failure to thrive**	**24 (20.2)**	**12 (10.1)**	**0.03**
**deafness**	**19 (16.0)**	**2 (1.7)**	**0.0001**	adenopathy	26 (21.8)	30 (25.2)	0.54
intellectual disabilities	6 (5.0)	4 (3.4)	0.52	immunodeficiency	6 (5.0)	2 (1.7)	0.28
Locomotor				hepatomegaly	3 (2.5)	4 (3.4)	1
arthralgia	73 (61.3)	80 (67.2)	0.34	splenomegaly	9 (7.6)	5 (4.2)	0.27
arthritis	42 (35.3)	29 (24.4)	0.07	recurrent infections	10 (8.4)	14 (11.8)	0.39
myalgia/myositis	32 (26.9)	35 (29.4)	0.67	**pharyngitis**	**14 (11.8)**	**26 (21.8)**	**0.04**
distorting arthropathy	5 (4.2)	3 (2.5)	0.72	polychondritis	2 (1.7)	0 (0.0)	0.50
osteitis	3 (2.5)	6 (5.0)	0.50	HLH	1 (0.8)	6 (5.0)	0.12
Renal				MDS	2 (1.7)	1 (0.8)	1
amyloidosis	6 (5.0)	2 (1.7)	0.28	obesity	4 (3.4)	4 (3.4)	1
kidney failure	10 (8.4)	4 (3.4)	0.10				
proteinuria	11 (9.2)	5 (4.2)	0.12				

^§^ Due to missing data, the number of cases and controls can differ from total group size (as *n*/known data (%)). * Genital aphthae are not displayed in this table as there were no patients with solely genital aphthae (without oral aphthae). To ascertain the count of patients with genital aphthae, refer to “bipolar aphthosis” in Table 2. Associations with a *p*-value < 0.05 are bolded. System categories are underligned. IBD: Inflammatory Bowl Disease, HS: Hidradenitis Suppurativa, HLH: Hemophagocytic Lympho Histiocytosis, MDS: Myelo Dysplasic Syndrome.

**Table 2 genes-14-01939-t002:** Characteristic combination at prescription.

Combinations	Cases *n* (%)	Controls *n* (%)	*p*
NO fever and/or NO CRP elevation	39 (32.8)	38 (31.9)	0.89
NO fever and NO CRP elevation	11 (9.2)	15 (12.6)	0.41
Bipolar aphthosis	8 (6.7)	17 (14.3)	0.06
**Hepatosplenomegaly**	**14 (11.8)**	**5 (4.2)**	**0.03**
No other system than inflammatory	1 (0.8)	2 (1.7)	1
At least 1 system (except inflammatory)	118 (99.2)	117 (98.3)	1
At least 2 systems (except inflammatory)	108 (90.9)	109 (91.6)	0.82
At least 3 systems (except inflammatory)	86 (72.3)	93 (78.2)	0.29
At least 4 systems (except inflammatory)	56 (47.1)	55 (46.2)	0.90
At least 5 systems (except inflammatory)	29 (24.4)	30 (25.2)	0.88
At least 6 systems (except inflammatory)	9 (7.6)	8 (6.7)	0.80
All 7 systems (except inflammatory)	2 (1.7)	0 (0.0)	0.50
Mean of the number of systems involved (except inflammatory), 95% CI	3.43 (0.51; 6.35)	3.47 (0.74; 6.20)	0.82

Associations with a *p*-value < 0.05 are bolded.

**Table 3 genes-14-01939-t003:** Diagnostic and predictive values linked to characteristics associated with panel positivity, along with fever and elevated CRP during crises.

Characteristics	Cases *n* (%)	Controls *n* (%)	OR (95% CI)	Se (%) (95% CI)	Sp (%) (95% CI)	PPV (%) * (95% CI)	NPV (%) * (95% CI)	Accuracy (%) * (95% CI)
Neonatal onset ^§^	12/104 (11.5)	1/107 (0.9)	13.83 (1.76; 108.38)	11.5 (6.1; 19.3)	**99.1** **(94.9; 100.0)**	48.2 (11.0; 87.5)	**93.7** **(93.3; 94.1)**	**92.9** **(88.6; 96.0)**
Early childhood onset ^§^	56/104 (53.8)	30/107 (28.0)	2.99 (1.69; 5.30)	53.9 (43.8; 63.7)	72.0 (62.5; 80.2)	12.6 (9.2; 17.1)	**95.4** **(94.2; 96.34)**	70.7 (64.1; 76.7)
Fever	86 (72.3)	87 (73.1)	0.96 (0.54; 1.70)	72.3 (63.3; 80.1)	26.9 (19.2; 35.8)	6.9 (6.0; 8.0)	**92.8** **(89.5; 95.12)**	30.1 (24.3; 36.30)
Crisis CRP elevation	102 (85.7)	98 (82.4)	1.29 (0.64; 2.58)	**85.7** **(78.1; 91.5)**	17.7 (11.3; 25.7)	7.3 (6.6; 8.1)	**94.3** **(90.1; 96.72)**	22.4 (17.3; 28.3)
Deafness	19 (16.0)	2 (1.7)	11.12 (2.53; 48.89)	16.0 (9.9; 23.8)	**98.3** **(94.1; 99.8)**	41.7 (14.6; 75.0)	**94.0** **(93.8; 94.4)**	**92.6** **(88.5; 95.6)**
Failure to thrive	24 (20.2)	12 (10.1)	2.25 (1.07; 4.75)	20.2 (13.4; 28.5)	**89.9** **(83.1; 94.7)**	13.1 (7.3; 22.3)	**93.7** **(93.1; 94.3)**	**85.0** **(79.9; 89.3)**
Pharyngitis	14 (11.8)	26 (21.8)	0.48 (0.24; 0.97)	11.8 (6.6; 19.0)	78.2 (69.7; 85.2)	3.9 (2.2; 6.9)	**92.2** **(91.3; 93.0)**	73.5 (67.4; 79.0)
Hepatosplenomegaly	14 (11.8)	5 (4.2)	3.04 (1.06; 8.73)	11.8 (6.6; 19.0)	**95.8** **(90.5; 98.6)**	17.4 (7.3; 36.2)	**93.5** **(93.0; 94.0)**	**89.9** **(85.4; 93.4)**

^§^ Due to missing data, the number of cases and controls can differ from total group size (as *n*/known data (%)). * These values are based on a panel positivity rate of 7%. Diagnostic and predictive values ≥ 85% are bolded. OR: Odds Ratio, Se: Sensitivity, Sp: Specificity, PPV: Positive Predictive Value, NPV: Negative Predictive Value.

**Table 4 genes-14-01939-t004:** Subgroup analyses.

Characteristics	32-Gene Panel Subgroup				55-Gene Panel Subgroup					62-Gene Panel Subgroup			
	Cases, *n* (%)	Controls, *n* (%)		*p*	Cases, *n* (%)		Controls, *n* (%)	*p*		Cases, *n* (%)	Controls, *n* (%)		*p*
Significantly associated variables													
neonatal onset ^§^	0/4 (0.0)	1/5 (20.0)		1	1/14 (7.1)		0/17 (0.0)	0.45		**8/49 (16.3)**	**0/49 (0.0)**		**0.006**
early childhood onset ^§^	2/4 (50.0)	2/5 (40.0)		1	7/14 (50.0)		7/17 (41.2)	0.62		**30/49 (61.2)**	**17/49 (34.7)**		**0.009**
deafness	1 (20.0)	0 (0.0)		1	5 (29.4)		1 (5.9)	0.17		**8 (14.0)**	**1 (1.8)**		**0.03**
failure to thrive	2 (40.0)	0 (0.0)		0.44	2 (11.8)		3 (17.6)	1		14 (24.6)	7 (12.3)		0.09
pharyngitis	0 (0.0)	2 (40.0)		0.44	3 (17.6)		4 (23.5)	1		**7 (12.3)**	**16 (28.1)**		**0.04**
hepatosplenomegaly	1 (20.0)	1 (20.0)		1	4 (23.5)		0 (0.0)	0.10		7 (12.3)	1 (1.8)		0.06
Trend variables													
childhood onset ^§^	4/4 (100.0)	4/5 (80.0)		1	13/14 (92.9)		17/17 (1.0)	0.26		45/49 (91.8)	40/49 (81.6)		0.14
adulthood onset ^§^	0/4 (0.0)	1/5 (20.0)		1	1/14 (7.1)		0/17 (0.0)	0.45		4/49 (8.2)	9/49 (18.4)		0.14
papillitis	1 (20.0)	0 (0.0)		1	2 (11.8)		0 (0.0)	0.48		1 (1.8)	0 (0.0)		1
arthritis	3 (60.0)	0 (0.0)		0.17	7 (41.2)		5 (29.4)	0,47		20 (35,1)	16 (28.1)		0.42
bipolar aphthosis	1 (20.0)	0 (0.0)		1	0 (0.0)		2 (11.8)	0.48		**1 (1.8)**	**8 (14.0)**		**0.03**
**Characteristics**	**114-Gene Panel Subgroup**						**302-Gene Panel Subgroup**						
	**Cases, *n* (%)**		**Controls, *n* (%)**			** *p* **	**Cases, *n* (%)**		**Controls, *n* (%)**			** *p* **	
Significantly associated variables													
neonatal onset ^§^	1/9 (11.1)		0/6 (0.0)			1	2/28 (7.1)		0/30 (0.0)			0.23	
early childhood onset ^§^	5/9 (55.6)		1/6 (16.7)			0.29	12/28 (42.9)		6/30 (20.0)			0.06	
deafness	1 (11.1)		0 (0.0)			1	4 (12.9)		0 (0.0)			0.11	
failure to thrive	1 (11.1)		0 (0.0)			1	5 (16.1)		2 (6.5)			0.42	
pharyngitis	1 (11.1)		1 (11.1)			1	3 (9.7)		3 (9.7)			1	
hepatosplenomegaly	0 (0.0)		2 (22.2)			0.47	2 (6.5)		2 (6.5)			1	
Trend variables													
childhood onset ^§^	8/9 (88.9)		4/6 (66.7)			0.53	24/28 (85.7)		22/30 (73.3)			0.24	
adulthood onset ^§^	1/9 (11.1)		2/6 (33.3)			0.53	4/28 (14.3)		8/30 (26.7)			0.24	
papillitis	0 (0.0)		0 (0;0)			1	0 (0.0)		0 (0.0)			1	
arthritis	3 (33.3)		1 (11.1)			0.58	9 (29.0)		7 (22.6)			0.56	
bipolar aphthosis	1 (11.1)		1 (11.1)			1	5 (16.1)		6 (19.4)			0.74	

^§^ Due to missing data, the number of cases and controls can differ from total subgroup size (as *n*/known data (%)). Associations with a *p*-value < 0.05 are bolded. Variables categories have been underligned.

**Table 5 genes-14-01939-t005:** Exploratory analyses: missing data exclusion.

Characteristics	Cases, *n* (%)	Controls, *n* (%)	OR (95% CI)	*p*-Value
Fever ^§^	86/110 (78.2)	84/116 (72.4)	1.37 (0.74; 2.51)	0.32
**Crisis CRP elevation ^§^**	**102/105 (97.1)**	**92/113 (81.4)**	**7.76 (2.24; 26.88)**	**0.002**

^§^ Due to missing data, the number of cases and controls differs from total group size (as *n*/known data (%)). Associations with a *p*-value < 0.05 are bolded. OR: Odds Ratio.

## Data Availability

Data are available upon request from the corresponding author.

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
