# Peer review of "Predictive Clinical and Biological Criteria for Gene Panel Positivity in Suspected Inherited Autoinflammatory Diseases: Insights from a Case–Control Study"

_genes, 2023, doi:10.3390/genes14101939_

Round 1
Reviewer 1 Report
Based on a case-control study, the article “Predictive Clinical and Biological Criteria for Gene Panel Positivity in Suspected Inherited Autoinflammatory Diseases: Insights from a Case-Control Study”. by Heiser et al. seeks to evaluate the performance of using clinical and biological criteria to predict gene panel testing positivity in patients with a suspected inherited genetic autoinflammatory disease.
However, the article has some shortcomings that need to be removed, or corrected and improved, please see below:
1. The title is not correctly written and contains the punctuation mark "period" or “.” at the end, which is NOT required or allowed in a title! Please, correct it!
2. The Abstract is too short and does not provide enough information about the study design, results, and conclusions: please, revise and improve the text.
3. There are abbreviations (CRP, NGS) in the Abstract that are not explained.
4. Some keywords are already in the title and should not have been included as keywords once again. Please, adopt the MeSH system for choosing the keywords and revise them accordingly.
5. At line 36, please correct as [2,3] instead of [2-3].
6. The academic style of the Introduction should be revised, and more references added to support the study.
7. All tables should be better explained and discussed.
8. There are drafting errors in the Tables, for example in Table 4, some characteristics are capitalized, while others are NOT
9. The final discussions should be improved; the conclusions also need to be reworded.
10. Please, at line 219: Similuk MN et al. [9], corrected as Similuk et al. [9],…
11. This article necessarily requires a graphical abstract for the reader to better understand the scientific content.
12. A Table of Abbreviations should be included at the end.
13. References must be checked and rewritten in the style required by the Genes journal and MDPI platform.
14. Academic editing of English, grammar, and style is required.
Overall, I recommend at least a minor revision.
I believe that after this revision provided by the authors on the issues suggested to be corrected and improved, it will provide useful and credible information for all readers, especially researchers and it is up to the Academic Editor to decide on its publication.
Thank you very much!
03 October 2023
Academic editing of English, grammar, and style is required.
Author Response
New version reviewer response
|
Response to Reviewer 1 Comments |
||
|
|
|
|
|
Thank you very much for taking the time to review this manuscript. Please find the detailed responses below and the corresponding revisions/corrections highlighted/in track changes in the re-submitted files.
|
||
|
Point-by-point response to Comments and Suggestions for Authors |
||
|
Comments 1: The title is not correctly written and contains the punctuation mark "period" or “.” at the end, which is NOT required or allowed in a title! Please, correct it! |
||
|
Response 1: This mistake was corrected. |
||
|
Comments 2: The Abstract is too short and does not provide enough information about the study design, results, and conclusions: please, revise and improve the text. |
||
|
Response 2: Agree. We have accordingly improved the Abstract. |
||
|
Comments 3: There are abbreviations (CRP, NGS) in the Abstract that are not explained. |
||
|
Response 3: You are right, The abbreviations for CRP and NGS were explained in the Abstract. |
||
|
Comments 4: Some keywords are already in the title and should not have been included as keywords once again. Please, adopt the MeSH system for choosing the keywords and revise them accordingly. |
||
|
Response 4: All the keywords were reworked adopting the MeSH system. |
||
|
Comments 5: At line 36, please correct as [2,3] instead of [2-3]. |
||
|
Response 5: Done |
||
|
Comments 6: The academic style of the Introduction should be revised, and more references added to support the study. |
||
|
Response 6: We improved the Introduction part and also with 2 new references. |
||
|
Comments 7: All tables should be better explained and discussed. |
||
|
Response 7: Clarification has been done notably about subgroup analyses. You could read theses modifications in track changes. Table 5 was added to represent exploratory analyses data. |
||
|
Comments 8: There are drafting errors in the Tables, for example in Table 4, some characteristics are capitalized, while others are NOT |
||
|
Response 8: These drafting errors have been corrected. All features are now uncapitalized in Table 4. |
||
|
Comments 9: The final discussions should be improved; the conclusions also need to be reworded. |
||
|
Response 9: The conclusion was rephrased as requested. We switched position of paragraph in the Discussion to improve the readability. |
||
|
Comments 10: Please, at line 219: Similuk MN et al. [9], corrected as Similuk et al. [9],… |
||
|
Response 10: Done |
||
|
Comments 11: This article necessarily requires a graphical abstract for the reader to better understand the scientific content. |
||
|
Response 11: We added a graphical abstract on the first page. |
||
|
Comments 12: A Table of Abbreviations should be included at the end. |
||
|
Response 12: At the conclusion of the article, preceding the references section, there is now an abbreviations table. |
||
|
Comments 13: References must be checked and rewritten in the style required by the Genes journal and MDPI platform. |
||
|
Response 13: We checked the Genes journal instruction and corrected the references. |
||
|
|
||
|
4. Response to Comments on the Quality of English Language |
||
|
Point 14: Academic editing of English, grammar, and style is required. |
||
|
Response 1: The article has been reviewed by the English edition services at CHU Montpellier |
||

Reviewer 2 Report
In the paper named “Predictive Clinical and Biological Criteria for Gene Panel Posi- 2 tivity in Suspected Inherited Autoinflammatory Diseases: In- 3 sights from a Case-Control Study” author make a case-control study in order to assess the clinical and biological criteria that predict gene panel positivity in 16 patients with a suspected inherited genetic autoinflammatory disease. As author state in the limitations the study is only an observational study however some good data can be obtained from the paper.
Only minor points are required:
1) None about the genes used in the different panels are said
2) The gene related in figure 1 that have a lot of variants were included in the different genes panels?
3) How author matched patients and controls in point 3.4
4) What about the missing values in point 3.5? What mean? Data not answered in the patient questioner
5) Results related in Point 3.5 are not present as a table please include the data
The English is fine, the paper is easy to read, only some complicated lines are present in the text. Only minor correction are necesary
Author Response
New version of reviewer response
|
Response to Reviewer 2 Comments |
||
|
|
|
|
|
Thank you very much for taking the time to review this manuscript. Please find the detailed responses below and the corresponding revisions/corrections highlighted/in track changes in the re-submitted files.
|
||
|
Point-by-point response to Comments and Suggestions for Authors |
||
|
Comments 1: None about the genes used in the different panels are said. |
||
|
Response 1: We thought it was too much (a maximum of 300 genes). This list can be found in our laboratory website https://umai-montpellier.fr/doc/Panel_MAI.pdf and in our former paper cited in [6]. |
||
|
Comments 2: The gene related in figure 1 that have a lot of variants were included in the different genes panels? |
||
|
Response 2: The genes depicted in Figure 1, whose variants were detected in positive panels, are genes that exist in at least one panel. Genes with the most variants are typically those that were discovered earliest, which often corresponds to the oldest genes in our genetic panel. It is crucial to note that there is only a single panel, which has evolved over time and has progressed in terms of the number of genes. I added a short explanation in the Introduction: “In our laboratory, we have developed a unique NGS panel that investigates inherited autoinflammatory diseases. The panel has undergone expansion in terms of the number of genes over time (32 to 302 genes).’’ |
||
|
Comments 3: How author matched patients and controls in point 3.4? |
||
|
Response 3: As described in Section 2.3 of the study design, the matching of cases and controls was based on the size of the panel, reflecting the analysis period. Consequently, there were an equal number of cases and controls for each panel version. To clarify, I have included this information in a new sentence: “Therefore cases and controls are matched on panel size (32, 55, 62, 114, 302 genes).’’ |
||
|
Comments 4: What about the missing values in point 3.5? What mean? Data not answered in the patient questioner |
||
|
Response 4: You are correct. The missing data correspond to the data that the prescriber did not indicate on the questionnaire. It should be noted that there is a difference in the main and exploratory analyses. In the main analysis, missing data are defined as negative for cases and positive for controls to intentionally underestimate sensitivity and specificity, as explained at the end of section 2.2. In the exploratory analysis, we excluded any missing data from the examination, which explains why group sizes are less than 119. |
||
|
Comments 5: Results related in Point 3.5 are not present as a table please include the data |
||
|
Response 5: Done |
||
|
|
||
|
Response to Comments on the Quality of English Language |
||
|
Point 1: The English is fine, the paper is easy to read, only some complicated lines are present in the text. Only minor corrections are necessary. |
||
|
Response 1: Typo were corrected. |
||
